# Functional Characterization of Fermented Beverages Based on Soy Milk and Sea Buckthorn Powder

**DOI:** 10.3390/microorganisms11061493

**Published:** 2023-06-04

**Authors:** Nicoleta-Maricica Maftei, Alina-Viorica Iancu, Alina Mihaela Elisei, Tudor Vladimir Gurau, Ana Yndira Ramos-Villarroel, Elena Lacramioara Lisa

**Affiliations:** 1Research Centre in the Medical-Pharmaceutical Field, Faculty of Medicine and Pharmacy University “Dunărea de Jos”, 800008 Galati, Romania; a_elisei@yahoo.com (A.M.E.); gtudorvladimir@gmail.com (T.V.G.); elena.lisa@ugal.ro (E.L.L.); 2Medical Laboratory Department, Clinical Hospital of Children Hospital “Sf. Ioan”, 800487 Galati, Romania; 3Medical Laboratory Department, Clinical Hospital of Infectious Diseases “Sf. Cuvioasa Parascheva”, 800179 Galati, Romania; 4School of Science of Agro and Environment, Campus the Guaritos, University of Oriente, Av. University, Maturín 6201, Venezuela; ay2170@gmail.com

**Keywords:** health, probiotic, *Bifidobacterium bifidus*, beverage, soy milk, sea buckthorn powder

## Abstract

Limitations of dairy products, such as lactose intolerance, problems related to a high cholesterol intake in diet, malabsorption, and the requirement for cold storage facilities, as well as an increasing demand for new foods and tastes, have initiated a trend in the development of non-dairy probiotic products. The possibility of producing beverages based on soy milk, sea buckthorn powder, and fermented by *Bifidobacterium bifidus* (*Bb-12^®^*, Bb) strain at different temperatures (30 °C and 37 °C) was examined. Strain viability, pH, and titratable acidity were measured during the fermentation period while the viability, pH, titratable acidity, and water holding capacity were determined during the storage time at 4 °C ± 1 °C within 14 days. Additionally, the survival and stability of *Bb-12^®^,* inoculated into a functional beverage when exposed to simulated gastrointestinal tract conditions, were assessed. The results obtained in this study revealed that the content of potent bioactive compounds in fermented soy milk and sea buckthorn powder depends on the processing conditions, the bacteria used in the fermentation step, and storage time.

## 1. Introduction

Currently, functional foods are the most promising in the field of nutrition. These foods are considered interesting from the consumer’s point of view for health and disease prevention because they use natural foods as part of a regular diet. The explanation for the increased interest in functional foods lies in the fact that these foods contain components with multiple health benefits, such as vitamins and minerals, prebiotics, and probiotics [1]. For centuries, traditionally, fermented foods and beverages have been produced in regions of Europe, Asia, America, and Africa using locally available raw materials [2]. One of the largest manufacturing sectors is the beverage industry, contributing to local economies in terms of providing value-added products but also a large workforce. Exports of food and beverages have doubled in European countries, obtaining receipts of more than EUR 90 billion, but they have also contributed to a balance of almost EUR 30 billion [3]. Fermented drinks and foods are obtained by spontaneous fermentation or by adding starter cultures or probiotics.

One of the great challenges of the food industry is the development of vegan products with improved nutritional, functional, and sensory characteristics, because in 2023, worldwide, the number of vegans will be approximately 88 million, which is around 1.1% of the world’s population [4]. For this reason, the scientific community is investigating natural and sustainable ingredients for the creation of innovative foods [5]. The functional beverage market is the fastest growing segment of the functional food sector. This market was valued at USD 25 billion in 2005 [6], and in the U.S., sales of functional beverages accounted for approximately 59% of the total market in 2012 [7]; by 2025, functional beverages are expected to represent 40% of total consumer demand [8]. Due to socio-cultural and socio-demographic factors, the differences between consumers and the acceptance of functional products have made the trends of the global market of functional drinks heterogeneous, and growth and development to be at different rates within and across countries [9].

A safe and cheap method of increasing the shelf life, nutritional, and organoleptic qualities of food is lactic acid fermentation [10]. The addition of active bacterial cultures can be considered a high-quality marker for functional foods. It is currently recognized that the consumption of probiotics has beneficial effects on the prevention of several diseases [11], and, in this context, many studies have demonstrated that probiotics can reduce blood pressure and hypertension [12], reduce symptoms associated with gastrointestinal disorders [13], help manage diabetes [14], and improve blood cholesterol levels [15], especially as metabolic syndrome associates obesity, dyslipidemia, hypertension, hyperglycemia, and the future risk of developing diabetes, cardiovascular complications, and stroke [16]. Likewise, probiotics reduce the risk of degenerative diseases [17], but they also prevent various allergies, and single-microbial-strain or multi-strain probiotics can serve to boost immunity against COVID-19 [18]. However, the minimum viable cell concentration, during consumption, for a probiotic food product has been estimated to be approximately 10^6^–10^7^ CFU/mL [19]. Typically, in the production of probiotic juices of non-lactic origin, the probiotics used are *Lactobacillus* and *Bifidobacteria* (*L. casei*, *L. paracasei*, *L. johnsonii*, *Lactobacillus acidophilus*, *L. helveticus*, *L. reuteri*, *L. delbrueckii* subsp. *bulgaricus*, *L. rhamnosus*, *B. bifidum*, *B. breve*, *B. lactis*, *B. longum*) but also other species, such as *Enterococcus faecium*, *Leuconostoc* spp., *Saccharomyces cerevisiae* var. *Boulardii*, *Weissella* spp., or mixed cultures of these microorganisms because they can provide different formulations in food matrices [20].

For healthy purposes, several raw materials can be used, soy being among those with the greatest potential [21]. In recent decades, soya (*Glycine max*) has been considered one of the cheapest and most important agricultural commodities due to its chemical composition; it has a high protein content (about 40%) compared to other leguminous species (20–30%) and cereals (8–15%) [22]. For this reason, interest in plant-based proteins is increasing, with them being used as substitutes for animal proteins [23], but also for their potential health benefits [24]. Nedele et al., 2021 [24] stated that soy-based products that include soy drink are considered a suitable substitute for milk, especially for vegans, people with allergies to milk proteins, and people with lactose intolerance. A good substrate for the growth of lactic acid bacteria, and especially *Bifidobacteria*, is soy milk [25]. In fact, soy milk is an aqueous extract from soybeans. It is considered an excellent source of dietary fiber, has a variety of micronutrients and phytochemicals, and a low fat content [26]. By fermenting soy milk, the aroma and the texture of the products is improved [25], soy milk being considered a good layer for the development and production of functional foods. It was also observed that the fermentation of soy milk with probiotic bacteria reduces the level of oligosaccharides and increases the level of free isoflavones [27].

Sea buckthorn (*Hippophae rhamnoides* L.) is a deciduous shrub, and according to research, it has lived on earth for approximately 200 million years [28]. Additionally, The World Health Organization has declared that sea buckthorn ranks first among the top ten raw materials used in health products [28]. Sea buckthorn fruits are a rich source of many nutrients and substances that promote health, such as polyphenols, carotenoids, tocopherols, ascorbic acid [29], and compounds belonging to the B vitamin complex, as well as minerals [30] and fatty acids [31]. It is also said that sea buckthorn berries provide a vast supply of compounds necessary for human nutrition because the content of multiple nutrients is superior compared to other common fruits (e.g., oranges or peaches) [30]. Due to the fact that, nowadays, as a rule, probiotic foods fall into the category of dairy products, which are unsuitable for people with lactose intolerance, it is appropriate to look for new formulations to expand the range of probiotic products with non-dairy foods. From this point of view, sea buckthorn is not sufficiently researched, but studies on other fruits have indicated the possibility of formulating probiotics from non-dairy products [30] (for example, probiotic juices of blueberries and blackberries [32], orange and pineapple juices [33], pomegranate juice [34], and sweet lemon [35]).

To better understand the action of biological compounds from sea buckthorn powder, the present study was conducted to investigate the effects of the addition of this powder on the microbiological and biochemical characteristics of probiotic soy beverages for possible use as a functional and lactose-free product. Thus, in this context, our study was designed to assess the functionality of *Bifidobacterium bifidus* when added to soy milk and sea buckthorn powder beverage matrices during fermentation, as well as during the storage period and gastrointestinal simulation.

## 2. Materials and Methods

### 2.1. Materials

#### 2.1.1. Soy Milk

The commercial soy beverage originated by Inedit Company, Romania, was purchased from a food store (Galati, Romania). The raw materials of the soy beverage were water and 95% soy from ecologic farming, with 1.9% lipids, 0.0% sugars, and 1.1% proteins. The soy milk product used was a sterilized product.

#### 2.1.2. Plant Material for Sea Buckthorn Powder

The sea buckthorn berries harvest period was September 2022 from the region of Moldavia (Romania). After reaching the laboratory, they were immediately stored in a freezer at −20 °C prior to the experiments. Mature and intact sea buckthorn was previously washed with distilled water to remove dust and surface impurities. After being washed, sea buckthorn berries were sorted and cleaned. Frozen sea buckthorn berries were thawed at room temperature for 12 h before they were squeezed with the help of a mixer. To obtain sea buckthorn powder, the collected juice was then freeze-dried with a Freeze-dryer Alpha 1–4 LDplus (Martin Christ Gefriertrocknungsanlagen GmbH, Osterode am Harz, Germany).

#### 2.1.3. Starter Culture (Probiotic Bacteria)

Probiotic lactic acid bacteria *Bifidobacterium bifidus* was used as a test organism for the fortification in the beverage matrices. The strain was procured from Christian Hansen (Hørsholm, Denmark) as a starter culture with the commercial name *Bb-12^®^*, *Bb*. The culture was maintained as frozen stocks in 50% glycerol and stored at −80 °C.

### 2.2. Fermentation for Production of Probiotic Fermented Beverage Based on Soy Milk and Sea Buckthorn Powder

To obtain functional beverage products, tests were carried out in the laboratory phase by fermenting soy milk and sea buckthorn powder (1% and 3% *w*/*v*) which were added in line with the variation in the fermentation temperature. After mixing the soy milk with sea buckthorn powder, the mixtures were homogenized separately with an Ultra Turrax blender (IKA, Merck, Germany) at 15,000 rpm until all the ingredients were dissolved in the soy milk. Then, 250 mL mixtures were inoculated with a starter culture and were fermented for 10 h at 30° and 37 °C. Samples were taken during the incubation at 0, 2, 4, 6, 8, and 10 h to test during and at the end of the fermentation; the samples were stored for 14 days at 4 ± 1 °C.

### 2.3. Analysis of Probiotic Fermented Beverage Based on Soy Milk and Sea Buckthorn Powder

#### 2.3.1. pH—Was Measured Using a pH Meter-MP2000 (Mettler Toledo, Greifensee, Switzerland)

For the titratable acidity assay, 10.0 mL of centrifuged beverage supernatant was titrated against 0.1 N NaOH using phenolphthalein as an indicator (pH 8.3). Titration was initiated by adding 2–3 drops of phenolphthalein to the sample, followed by the dropwise addition of 0.1 N NaOH with continuous swirling. The appearance of a stable light pink color indicated the point of neutrality, i.e., the end point of the titration. Titratable acidity was expressed in grams of lactic acid per 100 mL of the fermented product. The cell viability, pH, and titratable acidity of the samples were measured before and after the incubation period.

#### 2.3.2. Determination of Probiotic Viability

The counts of probiotic strain (*Bifidobacterium bifidus*) were conducted using the standard plate count method. Serial dilutions (with 0.9% NaCl solution) of fermented beverages were prepared. Aliquots of 1 mL of dilution were plated in MRS agar plates using the spread plate method. Then, the plates were incubated at 37 °C for 48 h under anaerobic conditions (Anaerocult^®^ A kit anaerobic incubator, Merck KGaA, Darmstadt, Germany). Plates containing from 25 to 250 colonies were chosen for enumeration, which was expressed in CFU/mL (colony-forming units per milliliter of product). Analyses were performed in triplicate.

### 2.4. In Vitro Survival Analysis for Simulated Gastrointestinal Conditions

In-vitro-simulated gastric juice (SGJ) was performed according to the procedure of the USP [36], national formulary: 2.0 g of NaCl, 3.2 g of pepsin, and 3.0 mL of concentrated HCl were diluted to 1 L and the pH was adjusted to 2.0 with concentrated HCl or sterile 0.1 mol L^−1^ NaOH.

Simulated intestinal juices (SIJs) were prepared by suspending pancreatin USP (P-1500) in sterile sodium chloride solution (0.5%, *w*/*v*) to a final concentration of 1 g L^−1^, with 4.5% bile salts (Oxoid, Merck, Germany) and adjusting the pH to 8.0 with sterile 0.1 mol L^−1^ NaOH.

For sterilization, both solutions were filtered through a 0.22 μm membrane. To replicate the peristaltic bowel movements and temperature of the human body, the experiment was carried out in a shaker incubator at 37 °C. Briefly, 0.2 mL from each type of beverage was taken and added in a flask and mixed with 10 mL of SGJ and incubated for 5, 30, 60, and 120 min and 60, 90, and 120 min, respectively, in the SIJs at 37 °C. Shaking was maintained at 50 rpm in each step. After the process was complete, the samples were evaluated for the viability of bacteria using pour plate techniques in MRS agar by anaerobic incubation at 37 °C for 3 days. The data are expressed as the means from three independent experiments with three replicates.

### 2.5. Water Holding Capacity (WHC)

The water holding capacity (WHC) was measured according to the method described by [37] with the same modification. Briefly, 10 g of beverage was transferred into a 20 mL glass tube and was centrifuged (2500 rpm for 10 min at 20 °C). The expelled water was removed and weighed. The percentage of the WHC was defined according to the equation:WHC = [(Sample weight − Expelled water)/Sample weight] × 100(1)

### 2.6. Statistical Analysis

All measurements and analyses were conducted on three prepared samples, and the results are presented as means ± standard deviations (SD). The data were analyzed using Statgraphics plus v.5.1 package (Manugistics Inc., Rockville, MA, USA) by a one-way analysis of variance, and a comparison of the means was conducted using Duncan’s multiple range test (DMRT) with significance levels of *p* < 0.05.

## 3. Results

### 3.1. Viable Cell Counts and Physicochemical Properties during Fermentation

To identify the effect of sea buckthorn powder on *Bifidobacterium bifidus* (Bb-12^®^) viability in fermented soy drinks, samples were analyzed at the beginning (time 0) and at the end of the fermentation process (10 h) using the plate method (Figure 1). At time 0, there were no significant differences in the drinks (*p* < 0.05), with a value of 4.3 × 10^6^ CFU·mL^−1^. The influence of sea buckthorn powder on *Bifidobacteria* could be observed after 10 h of fermentation at both temperatures (30° and 37 °C).

As the fermentation progressed, the number of CFU of *Bb-12*^®^ increased slightly in growth throughout the entire fermentation period to both temperatures of fermentation. At the end of the fermentation period, the number of viable cells of *Bb-12*^®^ was around 1.2 × 10^8^ CFU·mL^−1^.

Throughout fermentation, the pH decreased, suggesting that the fermentation proceeded normally and began the production of organic acids by bifidobacterial. All formulations exhibited a pH decrease (*p* < 0.05) throughout the fermentation period (Figure 2).

Figure 3 represents the changes in titratable acidity during the fermentation time. At the end of the fermentation process, the product had a titratable acidity of 0.8 g of lactic acid 100 mL^−1^ for the sample with 1% sea buckthorn powder and 1.14 g of lactic acid 100 mL^−1^ for the sample with 3% sea buckthorn powder for fermentation at 30 °C and 1.17 and 1.22 g of lactic acid 100 mL^−1^, respectively, for fermentation at 37 °C. The acidity changed significantly (*p* < 0.05) from all samples.

### 3.2. Changes in Cell Viability, pH, Titratable Acidity, and Water Holding Capacity during Storage

Figure 4 presents the changes in the cell viability of the *Bb-12*^®^ strain (*p* < 0.05) for the 14 days of cold storage at 4 ± 1 °C. This evaluation began after the 10 h fermentation process ended. The total viable cells of *Bb-12*^®^ were decreased moderately to 0.99 × 10^7^ for the beverage with 1% and 3% sea buckthorn powder, fermented at 30 °C, and 1.01 × 10^8^ and 1.05 × 10^8^ CFU·mL^−1^ for the beverage with 1% and 3% sea buckthorn powder, respectively, fermented at 37 °C at the end of the storage period. However, the functional beverage with 1% and 3% sea buckthorn powder fermented at 37 °C presented a bigger population of *Bb-12*^®^ compared to all the formulations (*p* < 0.05). After the storage period, the viability of *Bb-12*^®^ remained at an acceptable level (over 10^7^ CFU mL^−1^).

The changes in the pH values of functional samples during storage are shown in Figure 5. The pH values of the beverage from different treatments slightly significantly decreased (*p* < 0.05) during storage. The lowest pH value of 3.68 belonged to the beverage with 3% sea buckthorn which was fermented at 37 °C; however, in general, the pH values of all the beverages were around four. By contrast, the acidity values increased (*p* < 0.05) in all formulations after 14 days of the storage period. In Figure 6, the values for all beverages after their storage time are shown. The acidity values range between 1.05 and 1.29 g of lactic acid 100 mL^−1^ for the sample with 1% and 3% sea buckthorn powder, respectively, for fermentation at 30 °C. For fermentation at 37 °C, the values range between 1.25 and 1.38 g of lactic acid 100 mL^−1^ for the sample with 1% and 3% sea buckthorn powder, respectively.

Figure 7 illustrates the effects of different levels of sea buckthorn powder on the WHC. During the period of storage, the WHC was higher for all beverages. As shown in Figure 7, the WHC of the beverage with 3% sea buckthorn powder fermented at 37 °C was significantly higher than that of other groups (*p* < 0.05). As can be seen from Figure 7, the WHC of all beverages varied between 74 and 76% at the end of the refrigeration period.

### 3.3. Gastrointestinal Simulation

In order to follow the bio-accessibility of the *Bifidobacteria* in functional beverages, the samples were subjected to gastric and intestinal phases of the in vitro digestion protocol. Figure 8 shows the pre- and post-digestion counts of *Bb-12*^®^ in soy drinks, with 1% and 3% sea buckthorn powder fermented at 30 °C and 37 °C, respectively.

During the digestion process, the bacterial count in all the samples was reduced significantly (*p* < 0.05). There were lower reductions after the gastric simulation compared with the intestinal simulation for all samples. The positive effects of the concentration of sea buckthorn powder in the gastric simulation can be observed in the functional drink, where the soy drinks with 3% sea buckthorn powder were significantly higher (73.63% for sample fermented at 30 °C and 80.03% for sample fermented at 37 °C, respectively) than 1% sea buckthorn powder beverages (71.16% for sample fermented at 30 °C and 77.9% for sample fermented at 37 °C, respectively) (*p* < 0.05). The same positive effects of the concentration of sea buckthorn powder in the intestinal simulation for all samples can be observed in the functional beverages: 70.21% for sample fermented at 30 °C and 73.4% for sample fermented at 37 °C, respectively, for sample with 3% sea buckthorn powder compared with 67.2% for sample fermented at 30 °C and 70.9% for sample fermented at 37 °C, respectively, for sample with 1% sea buckthorn powder.

## 4. Discussion

During the development of non-dairy probiotic products, important attention should be granted at the formulation to the food matrix for probiotic delivery to obtain the highest efficiency. The beneficial impact of probiotics for health is not only due to the type of strain used but is also mostly dependent on the matrix environment. In conclusion, it is very important that the food matrix planned for use as a probiotic vehicle supports the high survival of probiotics for prolonged time periods. Hence, we have evaluated several in vivo and in vitro factors to assess the effect of the food matrix on *Bifidobacterium bifidus* (*Bb-12*^®^) strain. This study examined the possibility of producing a functional beverage based on soy milk, sea buckthorn powder in different concentrations, and fermented by the *Bb-12*^®^strain.

The cells viability of *Bb-12*^®^ was not affected by either the time, concentration of sea buckthorn powder, or the temperature. At the 10th h of fermentation, its number increased, and the results indicated that the presence of different components such as sugars and proteins in the beverage could have enhanced the growth by providing nutrition to the probiotic *Bifidobacteria* in all samples. The increase in cell numbers was higher at 37 °C for the beverages with 1 and 3% sea buckthorn powder compared to 30 °C. As shown in Figure 1, supplementation with different amounts of sea buckthorn powder in the beverage affects the growth of the *Bb-12^®^*strain; the cell numbers increased as the samples with the amount of sea buckthorn powder increased.

During storage, the cell numbers of the *Bb-12^®^* slightly decreased and remained around 1 × 10^7^ and 1 × 10^8^ CFU mL^−1^ for the beverage with 3% sea buckthorn powder fermented at 30 °C and 37 °C, respectively, (Figure 4). Generally, during storage, the cell numbers of *Bb-12*^®^ decreased slightly for all the beverages at both temperatures of fermentation. The analysis of variance for the probiotic counts showed that supplementation with different amounts of sea buckthorn powder in the beverage and different temperatures of fermentation affect the growth of *Bb-12*^®^. The cell numbers increased as the samples with the amount of sea buckthorn powder during fermentation increased and slightly decreased during the storage period. An increase in the maximum viability was observed in the sample with 3% sea buckthorn powder fermented at 37 °C, and a decrease in the minimum viability during the storage period was observed in the same beverage. Therefore, sea buckthorn powder and soy milk positively affected the development of *Bifidobacteria*, providing them with a favorable environment for growth. Inoguchi et al. [38] and Hara et al. [39] declared that soy milk contains soybean oligosaccharides that are utilized by *Bifidobacteria*. Additionally, [40] declared that the fermentation temperature is one of the important factors which affects the viability of probiotic microorganisms and other qualitative parameters of probiotic fermented products, and the favorable temperature for the growth of most probiotic strains is in the range of 37–43 °C. Our results comply with the study conducted by [41] where soy milk was fermented at two different temperatures (37 °C and 42 °C) using ABTS culture (*Lactobacillus acidophilus, Bifidobacterium* spp, *Streptococcus thermophilus*). They reported that *Bifidobacterium* grew better during fermentation at 42 °C than at 37 °C. Additionally, [42] reported that *Lactobacillus casei* grew significantly better at 37 °C than at 30 °C for 12 h of fermentation. The growth results obtained in this work for the analyzed *Bifidobacteria* are comparable to those reported by other authors for species and strains of another bacterial groups and soy milk [27,43,44,45]. Therefore, we can declare that it is very important to test the compatibility between probiotics, fermentation conditions, and the food matrix to provide a positive interaction capable of increasing the microbial viability during fermentation and the storage period.

The intended health benefits of probiotics can only be obtained when the food (dairy and non-dairy beverages) contains the required minimum viable microorganism count at the time of consumption. Generally, the food industry has declared that the minimum recommended level is 10^6^ CFU mL^−1^ at the time of consumption [46], but and US FDA has recommended that the minimum probiotic count in a probiotic food should be at least 10^6^ CFU mL^–1^. Knorr [47] declared that a daily intake of 10^8^–10^9^ probiotic, depending on the amount ingested and considering the effect of storage on probiotic viability, is essential to achieve probiotic benefits in the human organism. In our study, the viable cells of *Bb-12*^®^ remained at the level mentioned, which means that our beverage is a functional drink. Additionally, we can conclude that the beverage based on soy milk, sea buckthorn powder, and *Bb-12*^®^ could act as a trigger for new research to identify new balanced matrices.

Throughout fermentation, the pH values of all the samples decreased from the initial pH after 10 h of fermentation and increased in terms of cell count. All samples showed nearly the same final pH, around 3.90, except for the sample with 1% sea buckthorn powder which was fermented at 30 °C (pH = 4.13). As a result of the continuing metabolic activities of *Bifidobacteria* toward the end of fermentation, the pH lowered to around 3.7 for all samples, except for the sample with 1% sea buckthorn powder which was fermented at 30 °C (pH = 3.95). In accordance with our results, [48] declared that for a probiotic, *Prunus mume* puree had a final pH of 3.0 from an initial pH of 3.5 after 10 days of storage at room temperature. Additionally, insignificant changes in pH, from an initial pH of 5.5 to 5.7, were observed for a probiotic cheese-like product over a period of 4 weeks [49].

The production of organic acids during fermentation is linked to a decrease in the pH, as was also observed in this study, and a very low pH value increases the concentration of undissociated organic acids in fermented products, thereby enhancing the bactericidal effect of these acids [50]. Additionally, [50] suggested that beverages (such as fruit juices) with low pH values present a major challenge to probiotics. The optimum value of pH for the growth of *Bifidobacteria* is in the range of 6.0–7.0 [51]. Dunne et al. [52] declared that *Bifidobacteria* species are noted to be less acid-tolerant, and a pH level below 4.6 is against their survival. The minimal pH reduction in samples stored under refrigerated conditions can be attributed to the metabolic activity of *Bb-12*^®^.

The titratable acidity increased with a decrease in the pH for all functional beverages during fermentation for both temperatures. The sample with 1% sea buckthorn powder fermented at 30 °C had a lower value of titratable acidity compared with all the beverages during fermentation and the storage period. However, the titratable acidity increased as the pH decreased during fermentation for both temperatures and for all beverages. The titratable acidity increased, as has been revealed by several studies [53,54,55,56,57]. In this study, the increased acidity of the formulations can be the result of post-acidification because of continued fermentation by *Bb-12*^®^ during storage. The pH and titratable acidity values observed in our study can be indirect indicators of starter metabolic activity during storage, and they confirm our results regarding the viability of *Bb-12*^®^ and organic acid content observed during the refrigerated period. We can conclude that the acid tolerance of *Bb-12*^®^ depends upon the strain of the species and characteristics of the substrate. For example, [40] reported that *B. longum* survived best in the presence of acids and bile salts, and *B. lactis* in fermented milks. Additionally, Sheehan et al. [33] observed extensive differences in the acid resistance of *Lactobacillus* and *Bifidobacterium* when added to juices (orange, pineapple, and cranberry juices). All the strains survived better in orange and pineapple juices compared to cranberry juice [33].

Syneresis in the fermented soy milk decreased for both temperatures during storage. The WHC values obtained on the first day of sampling were higher than those found during storage, and they decreased with the decreasing sea buckthorn powder concentrations. The storage period did not significantly (*p* < 0.05) affect the WHC of the functional beverages. The results indicate that despite the increased acidification over the storage period, there was no tendency to release water by the functional beverage. Our results agree with the results reported by [58], on the contrary to the results observed by other authors who have studied fermented dairy beverages [59].

In our study, during the gastrointestinal simulation, a reduction in the total viable count of *Bb-12*^®^ in all the samples was observed. Considering the results previously presented in the literature, the survival of the SGJs of the *Bb-12*^®^ strain after 7 days of storage of fermented products has been analyzed. The drop in viable count throughout the passage from the stomach to the large and small intestine observed in this study is a major finding due to the extreme conditions, such as the low pH of stomach acid, HCl, and bile juice, encountered by the *Bb-12*^®^ which restrict probiotic activity in the gut. The results show that the sea buckthorn powder concentration and fermentation temperature had a significant effect on the survivability rate of *Bb-12*^®^ cells during exposure to acidic conditions. This finding indicates a protective effect of sea buckthorn powder on *Bb-12*^®^ cells and the possible migration of prebiotic constituents to the final product that stimulated survival through gastrointestinal simulation. Our results agree with the results reported by [60,61,62].

The viability of cells in the SIJs decreased easily after the first hour and then decreased gradually in the next hour. Chen et al. [63] suggested that bile salt can traverse the bacterial cell membrane which can cause bacterial death. These results agree with the results reported by [64].

Leahy et al. [65] declared that *Bifidobacterium* is a predominant member of the intestinal microbiota of humans, and it is also essential for maintaining the intestinal environment of the host. Likewise, *Bifidobacterium* exerts various beneficial effects such as in the synthesis of vitamins, those involved in the regulation of the intestinal environment, in the alleviation of constipation, in digestion and absorption, diarrhea, and infection, in enhancements to the immune system, in the amelioration of lactose intolerance, in the inhibition of carcinogenesis, in therapy and prophylaxis for inflammatory bowel disease, in reductions in serum cholesterol [65], and it has been used as a probiotic [66]; soybean oligosaccharides are also regarded as capable prebiotics and *Bifidobacterium* growth promoters [67]. Uriot et al. [68] suggests that the stomach is considered a biological and chemical barrier. Because of its low pH and the presence of proteolytic enzymes that promote an unfavorable environment to microorganisms, the difference in the pH in the intestine, combined with pancreatic fluids and bile salts, affects the chemical and stability of cell membranes (proteins and phospholipids). For this reason, this can cause cell disruption and homeostasis malfunction [68].

The results obtained in this study suggest the potential of soy milk and sea buckthorn powder as a functional food. From the point of view of the results obtained in this study, we can declare that the content of potent bioactive compounds in fermented soy milk and sea buckthorn powder depends on the processing conditions, the bacteria used in the fermentation step, and storage time. Finally, we wish to specify that it was very difficult to make comparisons to the literature because, presently, to the best of our knowledge, there have not been any more studies published about the behavior of the *Bb-12*^®^ strain when incorporated in a food matrix based on soy milk and sea buckthorn powder.

## 5. Conclusions

This research showed that the addition of 1 and 3% sea buckthorn powder in a fermented soy-milk-based product at 30 °C and 37 °C positively influences the growth and development of Bb-12^®^. The viability of these *Bifidobacteria* increased significantly after 10 h of fermentation, and the pH decreased at values of around 3.90. The titratable acidity increased with a decrease in the pH for all functional beverages during fermentation and the storage period for both temperatures, and the WHC values obtained decreased along with decreasing sea buckthorn powder concentrations. The *Bb-12*^®^ strain was able to survive in vitro in the human gastrointestinal tract, and the results obtained reveal that the *Bb-12*^®^ strain could be a potential probiotic which could be used in a soy-milk- and sea-buckthorn-powder-based beverage. According to the results obtained after digestion, the supportive impact of sea buckthorn powder depends on the bacteria used and temperature of fermentation. Thus, enriching soy beverages with sea buckthorn powder and fermenting with *Bb-12*^®^ presents a functional improved product. A further investigation into probiotics is required to understand the components responsible for the effective distribution of bacteria through the gastrointestinal tract, and the next step will be to verify whether these in vitro findings also apply to an in vivo situation because the potential probiotics must compete for mucosa receptors and nutrients with a plethora of intestinal microorganisms. Our research proves that there are opportunities to develop a new functional beverage and a market of functional foods based on fermented soy milk and sea buckthorn powder that incorporates health benefits.

## Figures and Tables

**Figure 1 microorganisms-11-01493-f001:**
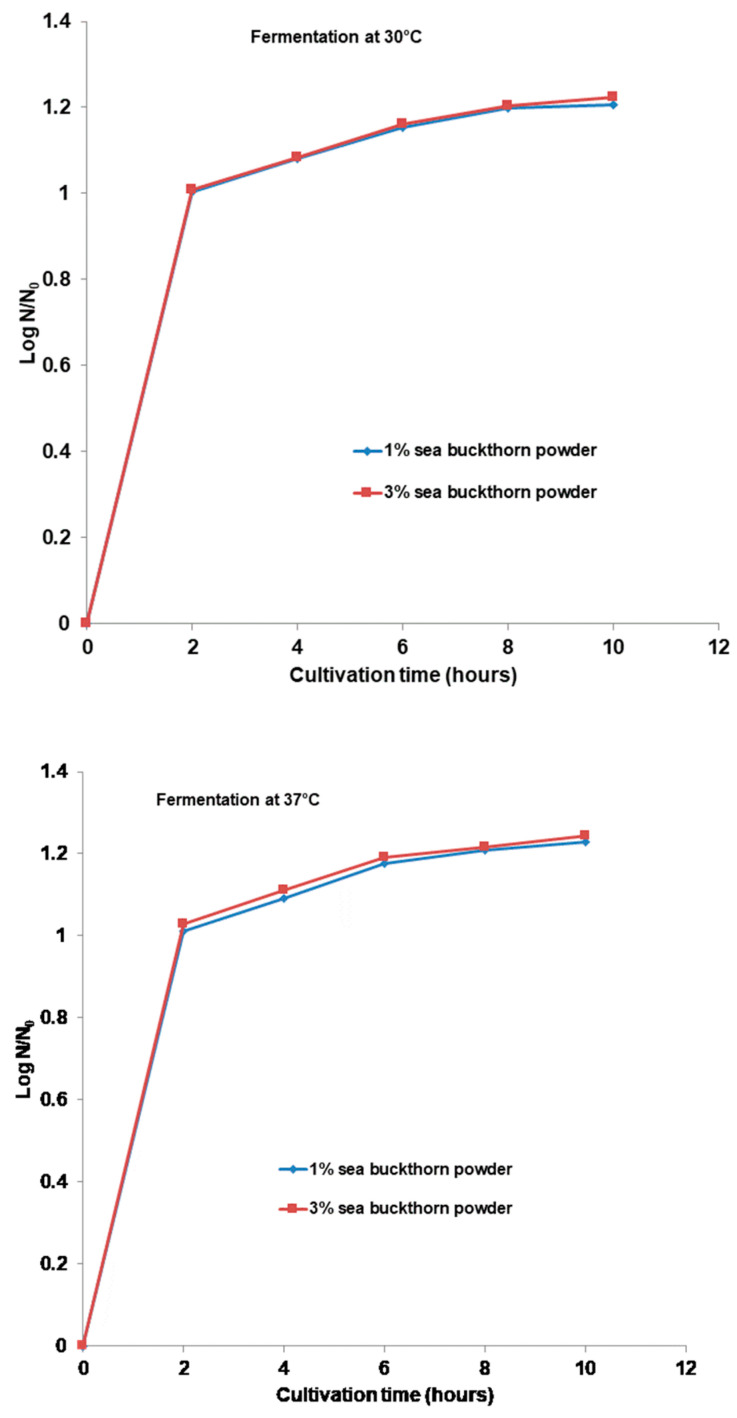
Cell viability profile of the fermentation with *Bifidobacteria*. Values for *Bb-12^®^* viable cell growth is displayed as mean values, CFU·mL^−1^, *n* = 3.

**Figure 2 microorganisms-11-01493-f002:**
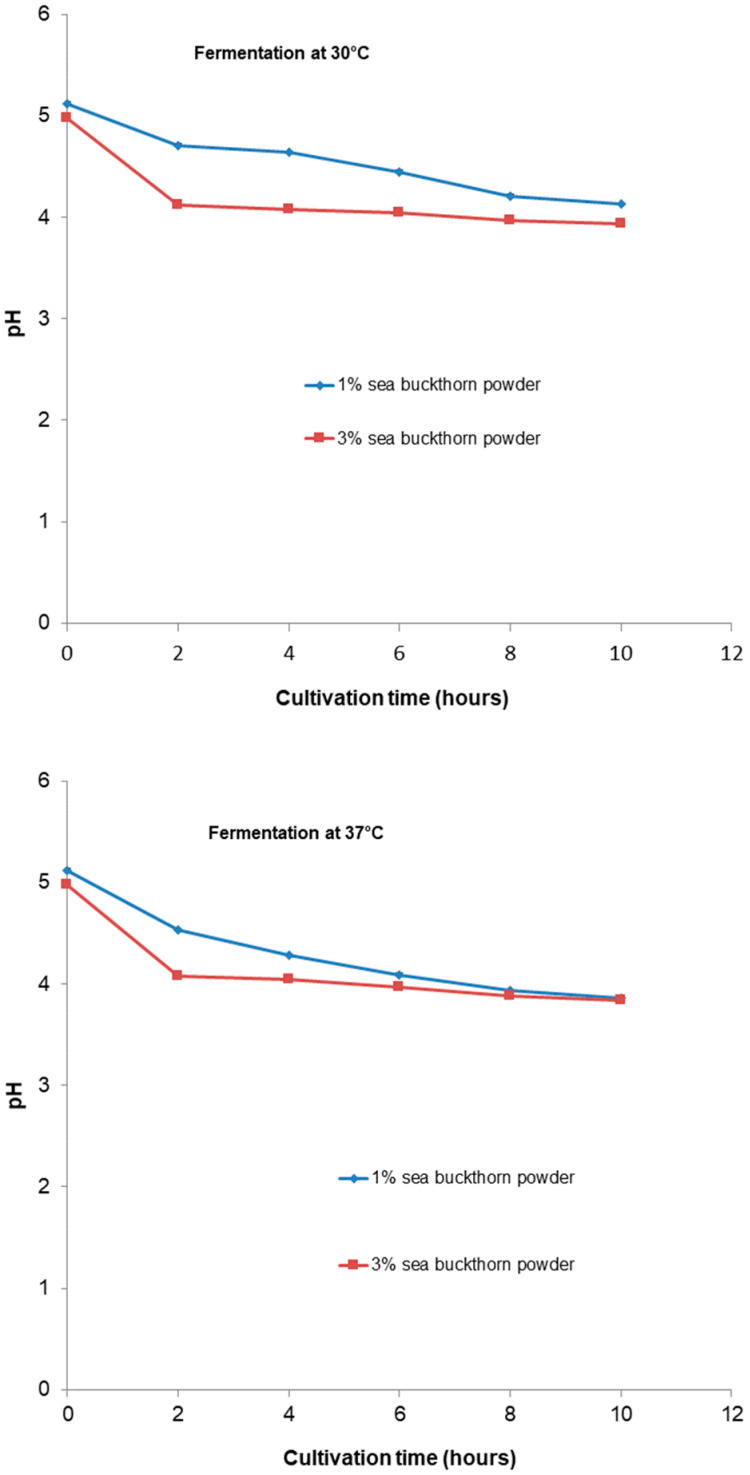
Changes in pH of functional beverages during fermentation period.

**Figure 3 microorganisms-11-01493-f003:**
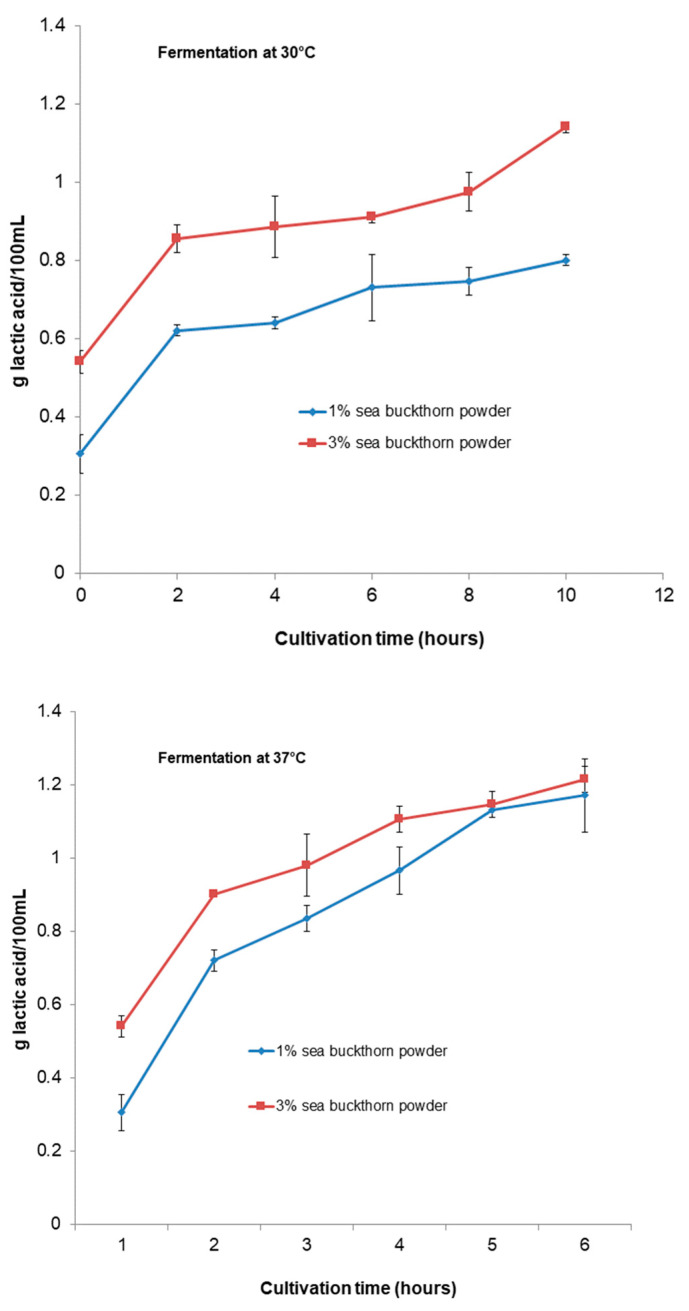
Changes in titratable acidity of functional beverages during fermentation period.

**Figure 4 microorganisms-11-01493-f004:**
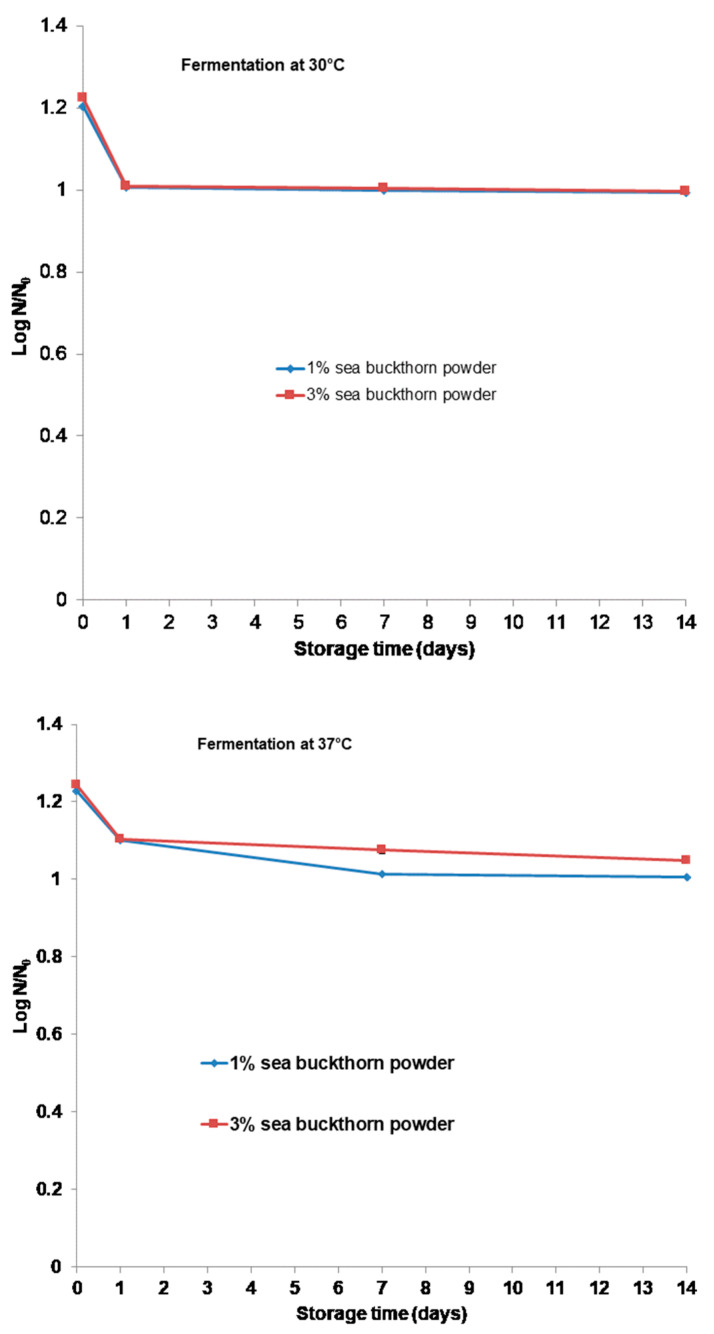
Survivability of probiotics in functional beverage during storage.

**Figure 5 microorganisms-11-01493-f005:**
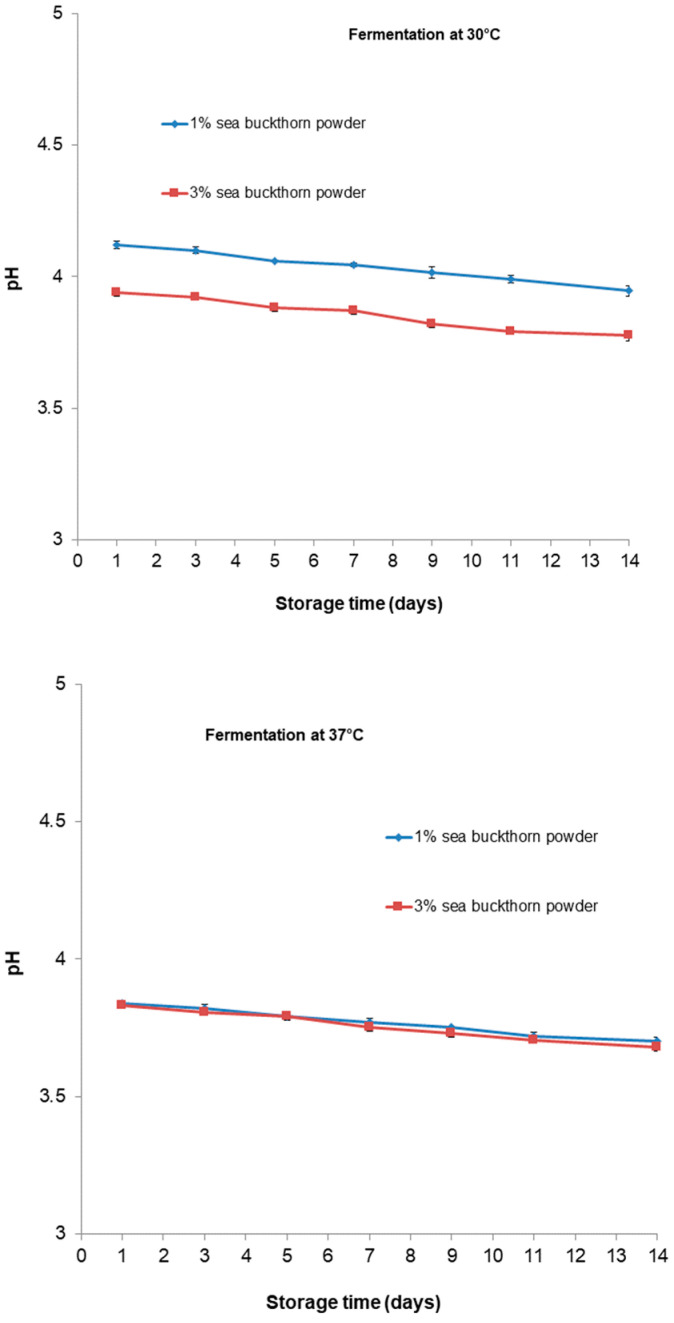
Changes in pH of functional beverages during storage period.

**Figure 6 microorganisms-11-01493-f006:**
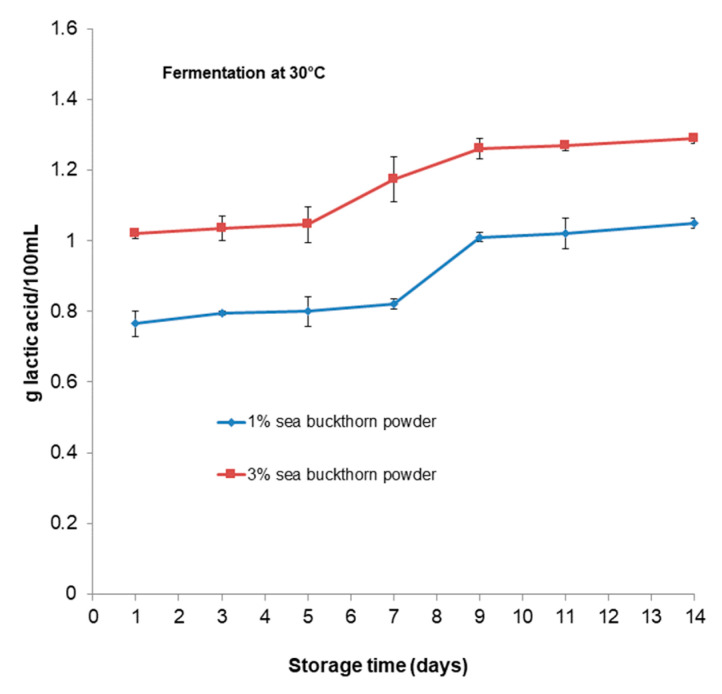
Changes in titratable acidity of functional beverages during storage period.

**Figure 7 microorganisms-11-01493-f007:**
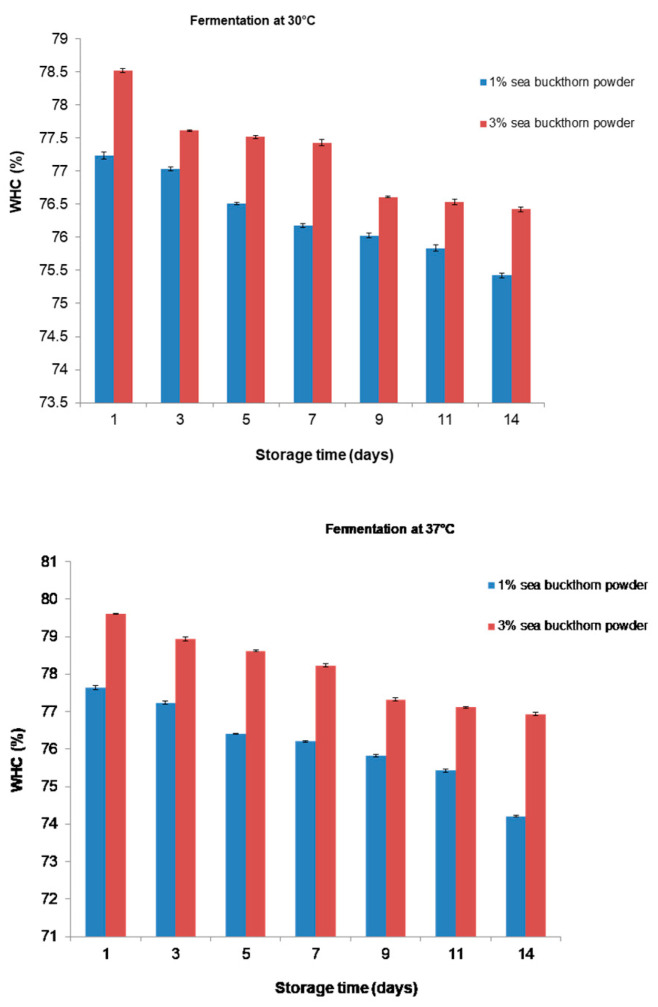
Changes in WHC of functional beverages during storage period.

**Figure 8 microorganisms-11-01493-f008:**
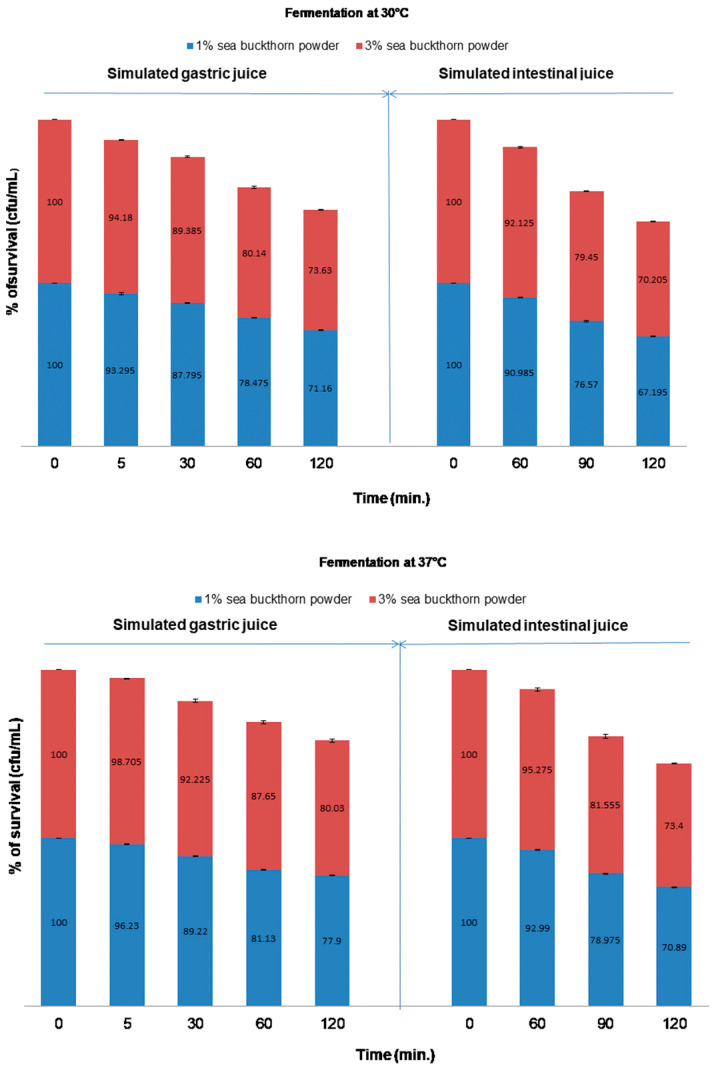
Survival rates of *Bb-12*^®^ incorporated into the beverages after continuous gastrointestinal simulations.

## Data Availability

Not applicable.

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
