# Peer review of "Functional Characterization of Fermented Beverages Based on Soy Milk and Sea Buckthorn Powder"

_microorganisms, 2023, doi:10.3390/microorganisms11061493_

Round 1

Reviewer 1 Report

This manuscript aims to develop a fermented beverage based on soymilk and sea buckthorn powder and assess its functional characteristics. However, only the probiotics characteristics of the beverage were evaluated, and nor the in vivo and in vitro fuctionality of BB12. In addition, the work lacks of the taste and smell characteristics, the basic as a beverage expecially a plant milk drink.  If the authors develop the drink for the purpose of not using dairy products because of their allergens, the fermented beverage based on soymilk also contain allergens. The experimental design need to be improved. 

The language is good.

Author Response

Response to Reviewer 1 Comments

Thank you for the time given to the review and we respond promptly to the comments, because we consider that all observations are objective and well founded.

Point 1: This manuscript aims to develop a fermented beverage based on soymilk and sea buckthorn powder and assess its functional characteristics. However, only the probiotics characteristics of the beverage were evaluated, and nor the in vivo and in vitro fuctionality of BB12. In addition, the work lacks of the taste and smell characteristics, the basic as a beverage expecially a plant milk drink. If the authors develop the drink for the purpose of not using dairy products because of their allergens, the fermented beverage based on soymilk also contain allergens. The experimental design need to be improved.

Response 1: Because until the present moment we did not find in the literature data refering to functional beverages based on soy and seabuckthorn powder, the scope of developing such a drink was:

  • to better understand the action of biological compounds from seabuckthorn powder due to the fact we have not worked with a standardized powder;
  • to investigate the effects of adding such powder on on microbiological and biochemical characteristics of probiotic soy beverages;
  • for a possible utilization as a functional lactose-free product which could be consumed by people with lactose intolerance ;

Also, when a new functional beverage is formulated or intended to be formulated, functionality and

safety of the active ingredient and/or product needs to be determined.

Refering to tests for senzorial evaluation, they were carried out but they were not described in this article because they have been studied multiple beverage variants (we have created a beverage with encapsulated probiotic and another microorganism) following that all of the results which present senzorial analisys to be described in a new article. As such, we have modified the scope of this study:„To better understand the action of biological compounds from sea buckthorn powder, the present study was conducted to investigate the effects of the addition of this powder on microbiological and biochemical characteristics of probiotic soy beverages for possible use as a functional and lactose-free product. Thus, in this context, our study was designed to assess functionality of Bifidobacterium bifidus when added to soy milk and sea buckthorn powder beverage matrices during fermentation, as well as during the storage period and gastrointestinal simulation“.

For all suggestions for corrections which were introduced in the attached word file, I have modified the text where it was suggested. All changes have been marked in red.

Reviewer 2 Report

Review for

 Functional characterization of fermented beverage based on soymilk and sea buckthorn powder

by

Nicoleta-Maricica Maftei, Alina-Viorica Iancu, Alina Mihaela Elisei, Tudor Vladimir Gurau, Ana Yndira,  Ramos-Villarroel, Elena Lacramioara Lisa

 Limitations of dairy products such as the presence of allergens and the requirement for  cold storage facilities, as well as an increasing demand for new foods and tastes have initiated a  trend in non-dairy probiotic product development

Fully true

---------------------

 The possibility of producing beverages based  on soy milk, sea buckthorn powder and fermented by Bifidobacterium bifidus (Bb-12®, Bb) strain at  different temperature (30°C and 37°C) was examined

Why such a strange combination?

soy milk, + sea buckthorn powder, fermented by Bifidobacterium bifidus (Bb-12®, Bb)

sea buckthorn powder: thousand tons produced in the world?

Please provide data by country, + names of industrial companies producing

Bifidobacterium bifidus (Bb-12®, Bb)

Why this specific strain, among so many….

---------------------------------------

 Also, survival and stability of Bb-12®, inoculated into a functional beverage when exposed to simulated gastrointestinal tract conditions, were assessed.

Strong debate about true value of some probiotics

https://www.nutraingredients.com/Article/2015/02/17/6-years-of-hurt-Probiotic-heavyweights-debate-the-EU-s-health-claim-blockade

https://www.nutraingredients.com/Article/2013/06/05/Ireland-Live-cultures-are-implied-probiotic-health-claims

current status?

---------------------------------

Plant material for sea buckthorn powder 104

Sea buckthorn berries harvest period: September 2022, from the region of Moldavia (Ro-105 mania). After reaching the laboratory, they were immediately stored in a freezer at -20°C 106 prior to the experiments. Mature and intact sea buckthorn was previously washed with 107 distilled water to remove dust and surface impurities. After being washed sea buckthorn 108 berries were sorted and cleaned. Frozen sea buckthorn berries were thawed at room tem-109 perature for 12 h before they were squeezed with the help of a mixer. To obtain sea buck-110 thorn powder, the collected juice was then freeze-dried with a Freeze-dryer Alpha 1-4 111 LDplus (Martin Christ Gefriertrocknungsanlagen GmbH).

Commercial product (standardized) should be used

---------------------------

Very bad quality for all figures   XY graphs  figures 1-6

Should be all redrawn

---------------------

Minor editing of English language required

Author Response

Response to Reviewer 2 Comments

Thank you for the time given to the review and we respond promptly to the comments, because we consider that all observations are objective and well founded.

Point 1: Limitations of dairy products such as the presence of allergens and the requirement for  cold storage facilities, as well as an increasing demand for new foods and tastes have initiated a  trend in non-dairy probiotic product development

Fully true

---------------------

Response 1: I have added new references in the text for your ideea.

Point 2: The possibility of producing beverages based  on soy milk, sea buckthorn powder and fermented by Bifidobacterium bifidus (Bb-12®, Bb) strain at  different temperature (30°C and 37°C) was examined

 Why such a strange combination?

 soy milk, + sea buckthorn powder, fermented by Bifidobacterium bifidus (Bb-12®, Bb)

sea buckthorn powder: thousand tons produced in the world?

Please provide data by country, + names of industrial companies producing

 Bifidobacterium bifidus (Bb-12®, Bb)

Why this specific strain, among so many….

---------------------------------------

Response 2: Reasons for choosing Bifidobacterium bifidus:

  1. Probiotics usually comprise bacteria, mainly Lactobacillus, Bacillus, and Bifidobacterium, Streptococcus, and Enterococcus, although some strains of yeast Saccharomyces genera have also been included in probiotic cultures. According to the International Scientific Association for Probiotics and Prebiotics (ISAPP) consensus panel, the probiotic mechanisms can be delivered by only a few strains of a particular class of bacteria, for example, Lactobacillus casei or Bifidobacterium bifidum [a]. A wide range of foods have been fermented or enriched in probiotics to be evaluated as possible carriers of these beneficial microorganisms and successfully placed on themarket. Several species of Lactobacillus and Bifidobacterium have become the most commonly used probiotic strains in these food products [b, c].
  2. Binda, S.; Hill, C.; Johansen, E.; Obis, D.; Pot, B.; Sanders, M.E.; Tremblay, A.; Ouwehand, A.C. Criteria to Qualify Microorganisms as “Probiotic” in Foods and Dietary Supplements. Front. Microbiol. 2020, 11, 1662
  3. Middelbos, I.; Fahey, G. Soybean Carbohydrates. In Soybeans: Chemistry, Production, Processing, and Utilization; AOCS Press:Urbana, IL, USA, 2008; pp. 269–296.
  4. Oliveira, A.S.; Niro, C.M.; Bresolin, J.D.; Soares, V.F.; Ferreira, M.D.; Sivieri, K.; Azeredo, H.M. Dehydrated strawberries for probiotic delivery: Influence of dehydration and probiotic incorporation methods. LWT 2021, 144, 111105.
  5. Because before this, we have done and published studies about soy milk, seabuckthorn syrup and used it as a microorganisms Lactobacillus casei paracasei and Bifidobacterium bifidus and have obtained satisfactory results, we have retaken up the studies utilinzing the combination soy milk, + sea buckthorn powder, fermented by Bifidobacterium bifidus (Bb-12®, Bb).

Nicoleta-Maricica Maftei, Rodica Dinica, Gabriela Bahrim. Functional Characterization of Fermented Beverage Based on Soymilk and Sea Buckthorn Syrup. The Annals of the University Dunarea de Jos–Food Technology. 2012; 36(1): 81-96.

Maftei N., Aprodu I., Dinică R., Bahrim G. New Fermented Functional Product Based on Soy Milk and Sea Buckthorn Syrup. Cyta-Journal of Food. 2013; Volume 11(Issue 3):256-269.

We also wanted to observe all the changes that appear in the product compared to the product based on soy milk and buckthorn syrup, in order to design a functional drink that is as good as possible from a biochemical, microbiological and safety point of view.

Point 3:   Also, survival and stability of Bb-12®, inoculated into a functional beverage when exposed to simulated gastrointestinal tract conditions, were assessed.

 Strong debate about true value of some probiotics

 https://www.nutraingredients.com/Article/2015/02/17/6-years-of-hurt-Probiotic-heavyweights-debate-the-EU-s-health-claim-blockade

https://www.nutraingredients.com/Article/2013/06/05/Ireland-Live-cultures-are-implied-probiotic-health-claims

current status?

---------------------------------

Response 3: When a new functional beverage is formulated or intended to be formulated, functionality and safety of the active ingredient and/or product needs to be determined. In the U.S. regulatory system, no separate category or set of regulations exist for functional food (FF). Instead, the regulations used for conventional foods applies to FF. To be marketed as a food, FF products are required to be safe and the ingredients are required to be approved as food additives or they must have the status of “generally recognized as safe” (Corbo et al., 2014).

In Europe, FFs are regulated by existing food legislation (Daliri &Lee, 2015, pp. 221–244) and a new food may require novel authorization and approval before the products are being launched into the market. Based on the safety assessment by the European Food Safety Authority (EFSA), the European Commission reviews proposals and decides to accept or reject a new food product (Corbo et al., 2014). European Commission Regulation 1924/2006 on “nutrition and health claims made on foods” regulates the nutrition and health claims in Europe (Corbo et al., 2014). In Europe, the procedure for the validation of a health claim is very lengthy and costly. Without taking into account the expenditures involved in providing scientific data to support applications to the European Food Safety Authority EFSA, the validation procedure of health claims costs from €4.51 to €7.65 million (Corbo et al., 2014; Nocella & Kennedy, 2012). Nutritional labelling is mandatory by the European Commission and food manufacturers are instructed to provide information about nutrients, energy value, and allergens present in food. European legislation has instructed that only those supplements (i.e. vitamins and minerals) that were on the positive list as of 1st August 2005 can be marketed, whereas new supplements must undergo a full safety assessment (Daliri & Lee, 2015, pp. 221–244; “Food Safety Authority of Ireland, 2011").

  1. Corbo, M. R., Bevilacqua, A., Petruzzi, L., Casanova, F. P., & Sinigaglia, M. (2014). Functional beverages: The emerging side of functional foods: Commercial trends, research, and health implications. Comprehensive Reviews in Food Science and Food Safety, 13(6), 1192–1206.
  2. Daliri, E. B. M., & Lee, B. H. (2015). Current trends and future perspectives on functional foods and nutraceuticals. In M. T. Liong (Vol. Ed.), Beneficial microorganisms in food and nutraceuticals. microbiology monographs: Vol. 27, (pp. 221–244). Cham: Springer.
  3. Food Safety Authority of Ireland (2011). Adopted from https://www.fsai.ie/uploadedFiles/Consol_Reg1169_2011.pdf.
  4. Nocella, G., & Kennedy, O. (2012). Food health claims–What consumers understand. Food Policy, 37(5), 571–580.

In our country the rules of the European Union are applied. Our team has conducted studies only on a laboratory level studies following at a pilot station.When we will have all of the results , we will analyze the registration of the beverage at OSIM (State Office for Inventions and Trademarks) to brevet it, keeping in mind that until this moment the product we created does not exist on market. Therefore, we consider that all the data mentioned above are not necessary in the article.

Point 4: Plant material for sea buckthorn powder 104

Sea buckthorn berries harvest period: September 2022, from the region of Moldavia (Romania). After reaching the laboratory, they were immediately stored in a freezer at -20°C prior to the experiments. Mature and intact sea buckthorn was previously washed with  distilled water to remove dust and surface impurities. After being washed sea buckthorn berries were sorted and cleaned. Frozen sea buckthorn berries were thawed at room temperature for 12 h before they were squeezed with the help of a mixer. To obtain sea buckthorn powder, the collected juice was then freeze-dried with a Freeze-dryer Alpha 1-4 LDplus (Martin Christ Gefriertrocknungsanlagen GmbH).

 Commercial product (standardized) should be used

---------------------------

Response 4: Reasons for choosing sea buckthorn powder made in our laboratory:

1) From the sea buckthorn fruit tree, we can see that the fresh sea buckthorn fruit is bright orange, and it will gradually oxidize during the production process of freeze-drying at minus 50 degrees, and the color remains intact!

Ordinary sea buckthorn fruit powder is made by high-temperature dehydration process, the antioxidant components in the sea buckthorn fruit are lost, and the fruit naturally turns dark yellow;

2) The sea buckthorn powder made by freeze-drying technology can make the fruit powder more delicate and fluffy. The ordinary sea buckthorn powder using high temperature technology has a small fineness, but the silty grinding degree is high, and the nutrition is completely destroyed;

3) The sea buckthorn powder produced by the low-temperature freeze-drying process can retain the nutrients and active substances in the sea buckthorn fruit to the greatest extent; ordinary sea buckthorn powder is made from fresh sea buckthorn fruit juice, filtration, concentration and spray drying. During the process, nutrients are lost and inactivated due to high temperature and oxidation;

4) The sea buckthorn powder produced by the low-temperature freeze-drying process can retain the nutrients and active substances in the sea buckthorn fruit to the greatest extent; ordinary sea buckthorn powder is made from fresh sea buckthorn fruit juice, filtration, concentration and spray drying. During the process, nutrients are lost and inactivated due to high temperature and oxidation;

5) Seabuckthorn freeze-dried powder has a smooth and non-granular taste. Ordinary sea buckthorn powder is grainy and tastes bad.

Point 5: Very bad quality for all figures XY graphs figures 1-6. Should be all redrawn

Response 5: I have improved the resolution of figures.

For all suggestions for corrections which were introduced in the attached word file, I have modified the text where it was suggested. All changes have been marked in red.

Response to Reviewer 2 Comments

Thank you for the time given to the review and we respond promptly to the comments, because we consider that all observations are objective and well founded.

Point 1: Limitations of dairy products such as the presence of allergens and the requirement for  cold storage facilities, as well as an increasing demand for new foods and tastes have initiated a  trend in non-dairy probiotic product development

Fully true

---------------------

Response 1: I have added new references in the text for your ideea.

Point 2: The possibility of producing beverages based  on soy milk, sea buckthorn powder and fermented by Bifidobacterium bifidus (Bb-12®, Bb) strain at  different temperature (30°C and 37°C) was examined

 Why such a strange combination?

 soy milk, + sea buckthorn powder, fermented by Bifidobacterium bifidus (Bb-12®, Bb)

sea buckthorn powder: thousand tons produced in the world?

Please provide data by country, + names of industrial companies producing

 Bifidobacterium bifidus (Bb-12®, Bb)

Why this specific strain, among so many….

---------------------------------------

Response 2: Reasons for choosing Bifidobacterium bifidus:

  1. Probiotics usually comprise bacteria, mainly Lactobacillus, Bacillus, and Bifidobacterium, Streptococcus, and Enterococcus, although some strains of yeast Saccharomyces genera have also been included in probiotic cultures. According to the International Scientific Association for Probiotics and Prebiotics (ISAPP) consensus panel, the probiotic mechanisms can be delivered by only a few strains of a particular class of bacteria, for example, Lactobacillus casei or Bifidobacterium bifidum [a]. A wide range of foods have been fermented or enriched in probiotics to be evaluated as possible carriers of these beneficial microorganisms and successfully placed on themarket. Several species of Lactobacillus and Bifidobacterium have become the most commonly used probiotic strains in these food products [b, c].
  2. Binda, S.; Hill, C.; Johansen, E.; Obis, D.; Pot, B.; Sanders, M.E.; Tremblay, A.; Ouwehand, A.C. Criteria to Qualify Microorganisms as “Probiotic” in Foods and Dietary Supplements. Front. Microbiol. 2020, 11, 1662
  3. Middelbos, I.; Fahey, G. Soybean Carbohydrates. In Soybeans: Chemistry, Production, Processing, and Utilization; AOCS Press:Urbana, IL, USA, 2008; pp. 269–296.
  4. Oliveira, A.S.; Niro, C.M.; Bresolin, J.D.; Soares, V.F.; Ferreira, M.D.; Sivieri, K.; Azeredo, H.M. Dehydrated strawberries for probiotic delivery: Influence of dehydration and probiotic incorporation methods. LWT 2021, 144, 111105.
  5. Because before this, we have done and published studies about soy milk, seabuckthorn syrup and used it as a microorganisms Lactobacillus casei paracasei and Bifidobacterium bifidus and have obtained satisfactory results, we have retaken up the studies utilinzing the combination soy milk, + sea buckthorn powder, fermented by Bifidobacterium bifidus (Bb-12®, Bb).

Nicoleta-Maricica Maftei, Rodica Dinica, Gabriela Bahrim. Functional Characterization of Fermented Beverage Based on Soymilk and Sea Buckthorn Syrup. The Annals of the University Dunarea de Jos–Food Technology. 2012; 36(1): 81-96.

Maftei N., Aprodu I., Dinică R., Bahrim G. New Fermented Functional Product Based on Soy Milk and Sea Buckthorn Syrup. Cyta-Journal of Food. 2013; Volume 11(Issue 3):256-269.

We also wanted to observe all the changes that appear in the product compared to the product based on soy milk and buckthorn syrup, in order to design a functional drink that is as good as possible from a biochemical, microbiological and safety point of view.

Point 3:   Also, survival and stability of Bb-12®, inoculated into a functional beverage when exposed to simulated gastrointestinal tract conditions, were assessed.

 Strong debate about true value of some probiotics

 https://www.nutraingredients.com/Article/2015/02/17/6-years-of-hurt-Probiotic-heavyweights-debate-the-EU-s-health-claim-blockade

https://www.nutraingredients.com/Article/2013/06/05/Ireland-Live-cultures-are-implied-probiotic-health-claims

current status?

---------------------------------

Response 3: When a new functional beverage is formulated or intended to be formulated, functionality and safety of the active ingredient and/or product needs to be determined. In the U.S. regulatory system, no separate category or set of regulations exist for functional food (FF). Instead, the regulations used for conventional foods applies to FF. To be marketed as a food, FF products are required to be safe and the ingredients are required to be approved as food additives or they must have the status of “generally recognized as safe” (Corbo et al., 2014).

In Europe, FFs are regulated by existing food legislation (Daliri &Lee, 2015, pp. 221–244) and a new food may require novel authorization and approval before the products are being launched into the market. Based on the safety assessment by the European Food Safety Authority (EFSA), the European Commission reviews proposals and decides to accept or reject a new food product (Corbo et al., 2014). European Commission Regulation 1924/2006 on “nutrition and health claims made on foods” regulates the nutrition and health claims in Europe (Corbo et al., 2014). In Europe, the procedure for the validation of a health claim is very lengthy and costly. Without taking into account the expenditures involved in providing scientific data to support applications to the European Food Safety Authority EFSA, the validation procedure of health claims costs from €4.51 to €7.65 million (Corbo et al., 2014; Nocella & Kennedy, 2012). Nutritional labelling is mandatory by the European Commission and food manufacturers are instructed to provide information about nutrients, energy value, and allergens present in food. European legislation has instructed that only those supplements (i.e. vitamins and minerals) that were on the positive list as of 1st August 2005 can be marketed, whereas new supplements must undergo a full safety assessment (Daliri & Lee, 2015, pp. 221–244; “Food Safety Authority of Ireland, 2011").

  1. Corbo, M. R., Bevilacqua, A., Petruzzi, L., Casanova, F. P., & Sinigaglia, M. (2014). Functional beverages: The emerging side of functional foods: Commercial trends, research, and health implications. Comprehensive Reviews in Food Science and Food Safety, 13(6), 1192–1206.
  2. Daliri, E. B. M., & Lee, B. H. (2015). Current trends and future perspectives on functional foods and nutraceuticals. In M. T. Liong (Vol. Ed.), Beneficial microorganisms in food and nutraceuticals. microbiology monographs: Vol. 27, (pp. 221–244). Cham: Springer.
  3. Food Safety Authority of Ireland (2011). Adopted from https://www.fsai.ie/uploadedFiles/Consol_Reg1169_2011.pdf.
  4. Nocella, G., & Kennedy, O. (2012). Food health claims–What consumers understand. Food Policy, 37(5), 571–580.

In our country the rules of the European Union are applied. Our team has conducted studies only on a laboratory level studies following at a pilot station.When we will have all of the results , we will analyze the registration of the beverage at OSIM (State Office for Inventions and Trademarks) to brevet it, keeping in mind that until this moment the product we created does not exist on market. Therefore, we consider that all the data mentioned above are not necessary in the article.

Point 4: Plant material for sea buckthorn powder 104

Sea buckthorn berries harvest period: September 2022, from the region of Moldavia (Romania). After reaching the laboratory, they were immediately stored in a freezer at -20°C prior to the experiments. Mature and intact sea buckthorn was previously washed with  distilled water to remove dust and surface impurities. After being washed sea buckthorn berries were sorted and cleaned. Frozen sea buckthorn berries were thawed at room temperature for 12 h before they were squeezed with the help of a mixer. To obtain sea buckthorn powder, the collected juice was then freeze-dried with a Freeze-dryer Alpha 1-4 LDplus (Martin Christ Gefriertrocknungsanlagen GmbH).

 Commercial product (standardized) should be used

---------------------------

Response 4: Reasons for choosing sea buckthorn powder made in our laboratory:

1) From the sea buckthorn fruit tree, we can see that the fresh sea buckthorn fruit is bright orange, and it will gradually oxidize during the production process of freeze-drying at minus 50 degrees, and the color remains intact!

Ordinary sea buckthorn fruit powder is made by high-temperature dehydration process, the antioxidant components in the sea buckthorn fruit are lost, and the fruit naturally turns dark yellow;

2) The sea buckthorn powder made by freeze-drying technology can make the fruit powder more delicate and fluffy. The ordinary sea buckthorn powder using high temperature technology has a small fineness, but the silty grinding degree is high, and the nutrition is completely destroyed;

3) The sea buckthorn powder produced by the low-temperature freeze-drying process can retain the nutrients and active substances in the sea buckthorn fruit to the greatest extent; ordinary sea buckthorn powder is made from fresh sea buckthorn fruit juice, filtration, concentration and spray drying. During the process, nutrients are lost and inactivated due to high temperature and oxidation;

4) The sea buckthorn powder produced by the low-temperature freeze-drying process can retain the nutrients and active substances in the sea buckthorn fruit to the greatest extent; ordinary sea buckthorn powder is made from fresh sea buckthorn fruit juice, filtration, concentration and spray drying. During the process, nutrients are lost and inactivated due to high temperature and oxidation;

5) Seabuckthorn freeze-dried powder has a smooth and non-granular taste. Ordinary sea buckthorn powder is grainy and tastes bad.

Point 5: Very bad quality for all figures XY graphs figures 1-6. Should be all redrawn

Response 5: I have improved the resolution of figures.

For all suggestions for corrections which were introduced in the attached word file, I have modified the text where it was suggested. All changes have been marked in red.

Round 2

Reviewer 1 Report

The first sentence of ABSTRACT about the limitation  of dairy products because of presence of allergens is not legitimate based on the research objective, which can mislead the consumers.  Lactose intolerance may be appropriate.

Author Response

Thank you again for the time given to the review and we respond promptly to the comments, because we consider that all observations are objective and well founded.

Point 1: The first sentence of ABSTRACT about the limitation of dairy products because of presence of allergens is not legitimate based on the research objective, which can mislead the consumers.  Lactose intolerance may be appropriate.

Response 1: The first sentence of the ABSTRACT was changed as per your suggestion.

For all suggestions for corrections which were introduced in the attached word file, I have modified the text where it was suggested. All changes have been marked in purple.

Reviewer 2 Report

Revision is OK

Only one comment:

""because currently 79 million people follow a vegan diet [4].""

this data appears as very low at a worldwide level....

Author Response

Thank you again for the time given to the review and we respond promptly to the comments, because we consider that all observations are objective and well founded.

Point 1:"because currently 79 million people follow a vegan diet [4].""

this data appears as very low at a worldwide level....

Response 1: I have added new references in the text.

For all suggestions for corrections which were introduced in the attached word file, I have modified the text where it was suggested. All changes have been marked in purple.
